# Low-Risk Planned Out-of-Hospital Births: Characteristics and Perinatal Outcomes in Different Italian Birth Settings

**DOI:** 10.3390/ijerph17082718

**Published:** 2020-04-15

**Authors:** Marta Campiotti, Rita Campi, Michele Zanetti, Paola Olivieri, Alice Faggianelli, Maurizio Bonati

**Affiliations:** 1Associazione Nazionale Culturale Ostetriche Parto a Domicilio e Casa Maternità, 21056 Induno Olona, VA, Italy; martacampiotti1@gmail.com (M.C.); ostetricaolivieri@hotmail.com (P.O.); faggianelli@libero.it (A.F.); 2Laboratory for Mother and Child Health, Department of Public Health, Istituto di Ricerche Farmacologiche Mario Negri IRCCS, Via Negri M. 2, 20156 Milan, Italy; rita.campi@marionegri.it (R.C.); michele.zanetti@marionegri.it (M.Z.)

**Keywords:** out-of-hospital birth, home delivery, delivery practices, planned birth, midwifery, Italy

## Abstract

Background: The present observational study aimed to describe women and delivery characteristics and early birth outcomes according to planned out-of-hospital delivery and to compare this information with comparable planned in-hospital deliveries. Methods: 1099 healthy low-risk women who delivered out-of-hospital between 2014 to 2018, with a gestational age of 37–42 completed weeks of pregnancy, with single, vertex babies whose birth was expected to be vaginal and spontaneous were enrolled. Moreover, a case-control study was designed comparing characteristics of these births to a matched 1:5 sample. Results: living in a medium city (RR 1.81, 95% CI 1.19–2.74), being multiparous (RR 1.66, CI 1.09–2.51), having the first child at ≥35 years old (RR 1.84, CI 1.02–3.33), not working (RR 1.77, CI 1.06–2.96), not being omnivorous (RR 1.80, CI 1.08–3.00), and not smoking (RR 2.53, CI 1.06–6.07) were all related to an increased chance of delivering at home compared to in a freestanding midwifery unit. The significant factors in choosing to give birth out-of-hospital instead of in-hospital were living in a large or medium city (OR 2.20; 1.75–2.77; OR 2.41; 1.93–3.02) and having a secondary or higher level of education (OR > 2 for both parents). Within the first week of delivery, 6 of 1099 mothers and 19 of 1099 neonates were hospitalized. Conclusions: out-of-hospital births in women with low-risk pregnancies is a possible option that needs to be planned, monitored, regulated, and evaluated according to healthcare control systems in order to work, as in hospitals, for the safest and most effective care to a mother and her neonate(s).

## 1. Introduction

After the second half of the past century in high resource countries, home births gave way to hospital births [1] and this probably led to the observed decrease in the rate of perinatal, neonatal, and maternal mortality [2]. This was one of the possible causes, but improvements in living conditions and general health at that time were also relevant. Dependence on technology also increased and diminished confidence in women’s innate ability to give birth without intervention [3]. The issue of out-of-hospital birth has re-emerged, the discussion about pro and cons of planned home birth continues [4], the time for greater collaboration across models of care is repeated [5], indications by professional organizations and societies are also produced [6,7,8,9], but little research has been undertaken to produce evidence on the safety of such births [10,11,12,13].

In resource rich countries, rates of planned out-of-hospital births (i.e., at home or in birth units) are low, but vary widely, and are highest in the Netherlands. Here, women can choose birth centers (centers with homelike surroundings), and 11.4% of births take place there and 16.3% at home [14,15]. In New Zealand, the overall rate of home births is around 3–5% of all recorded births [16], whereas in Japan, the rate is around 1.1% [17]. In Wales, England, Scotland, Iceland, and Switzerland, out-of-hospital rates are just 1–3%, while in other European countries, rates are even less at 1% [15]. In contrast, in Australia, only 0.3% of all births occur at home [18]. In the US, the rate has risen recently but is about 1.5% [19]. There are no official detailed statistics in Italy about these births, but the rate is reported to be around 0.004–0.01% (287–545 out-of-hospital births rather than all causes in the 2008–2015 period) [20]. Variation in the level of integration into the health care system across different settings is a further important factor comparing prevalence and outcomes of home birth between studies [21]. The main aim of this study was to explore, for the first time in Italy, out-of-hospital births and identify variables that predict them. There were two specific objectives and steps of the study: (1) to describe the women, delivery characteristics, and early birth outcomes for births planned out-of-hospital, including the comparison of home deliveries with those in freestanding midwifery units; (2) to compare the characteristics of women giving birth out-of-hospital to those of women who have planned hospital births.

## 2. Materials and Methods

### 2.1. Study Design

Our intent was twofold: to describe with an observational study the characteristics of women who planned to deliver out-of-hospital and the early birth outcomes. For the second objective, we used the same data in a case-control study comparing the characteristics of planned out-of-hospital births to those of planned hospital births.

### 2.2. Patient and Public Involvement

There was no patient or public involvement in this study.

### 2.3. Setting

The National Association of Out-of-Hospital Midwives (Associazione Nazionale Culturale Ostetriche Parto a Domicilio e Casa Maternità), founded in 1981, is a network of qualified midwives who provide care and support to pregnant women and their babies in the perinatal period. Almost all midwives who assist out-of-hospital births belong to this association. These midwives, with university level education and national accreditation standards, work privately outside the National Health System. They assist births at home or in private freestanding units run by midwives (a midwifery unit) and only for women who meet the criteria for such births as laid out in the national and international guidelines [9,22,23]. 

### 2.4. Sample

Women cared for by the out of hospital midwives association constituted the target population [24]. They were healthy low-risk women, with a gestational age of 37 to 42 completed weeks of pregnancy, with single, vertex babies whose birth was expected to be vaginal and spontaneous and who agreed to be transferred to hospital care if problems occurred. All out-of-hospital births from 2014 to 2018 reported in the association’s registry data were extracted. Data on 1138 women who planned out-of-hospital delivery (home births or in a freestanding midwifery unit) were analyzed. Data on about 39 women (3.4%) who began with out-of-hospital care but gave birth in hospital (3 for personal decision, 12 for fetal mal presentation, 18 for post-term pregnancy, and 6 hypertension) were not included in the present study. A total of 1099 out-of-hospital births constituted the sample. Data on maternal and paternal characteristics and birth related characteristics were analyzed.

A case-control study was performed to compare women who gave birth out of the hospital with women who gave birth in the hospital. For the case-control study, the Certificate of Delivery Assistance database (CedAP) of the Lombardy Region, which contains information on inpatient deliveries provided by any hospital or clinic included in the Regional Health System, was used to form the control group. The entire sample of 1099 women who delivered out-of-hospital formed the study group (cases).

The control group was extracted (1:5) and comprised a random sample of 5495 Italian women who gave birth in the hospital, with the same characteristics for pregnancy and delivery of the target population: healthy low-risk women with a gestational age of 37 to 42 completed weeks of pregnancy, with single, vertex babies, whose birth was vaginal and spontaneous, and delivered the same day. The control group was matched for maternal age, as well as for residential area (municipality) when possible, and gestational age, with the same inclusion and exclusion criteria for the cases.

### 2.5. Statistical Analyses

Categorical variables were summarized using proportions and associations tested using chi-square or Fisher’s exact test where applicable. Continuous variables were summarized using means and standard deviations for normally distributed data, while skewed data were summarized using medians. One-way ANOVA (F-value) was used to test difference of means for normally distributed continuous variables and the Mann–Whitney U test for skewed continuous variables. In the bivariable analyses, to identify risk factors associated with delivering at home or in a freestanding midwifery unit, we computed relative risks (RR) considering the significance of the confidence intervals. Statistical significance was evaluated using 95% confidence interval and a two-tailed *p*-value of <0.05. Absolute (marginal) differences were also calculated and reported in Appendix A. In the multivariable analysis, a log-binomial regression model was used.

For the second step of the study, odds ratios (OR) were calculated between women with out-of-hospital births and women with hospital births for the different categories of the explanatory variables through bivariate analyses. A 95% confidence interval for the odds ratio was calculated with a variance described by Mantel and Haentzel [25]. In addition to the multivariate analysis, conditional logistic regression analysis was performed. Women with an out-of-hospital birth were compared with women having a hospital birth. Sensitivity analysis was performed by running two separate models, adding confounders with missing values (residential area, number of children, and occupational status).

All data management and analyses were performed with the use of SAS software, version 9.4 (SAS, Institute Inc., Cary, NC, USA).

## 3. Results

Data on 1099 Italian women who delivered out-of-hospital were collected, 848 (77%) of the births were at home, and 251 (23%) at a midwifery unit. Most of the births (71.9%) took place in the North. The average age of the mothers and fathers was 34.0 ± 4.7 years (mean ± standard deviation) and 37.1 ± 6.0, respectively (Table 1). The mothers had a higher level of education than fathers (*p* ≤ 0.001), and the fathers worked more often (98.4%) than mothers (78.9%) (*p* ≤ 0.001). First, the distribution of mothers for number of children, age at delivery, occupational status, smoking, and educational level was different between mothers who delivered at home and in a midwifery unit.

In the multivariate regression model, living in a medium city (RR 1.81, 95% CI 1.19–2.74), being multiparous (RR 1.66, CI 1.09–2.51), having the first child at ≥35 years old (RR 1.84, CI 1.02–3.33), not working (RR 1.77, CI 1.06–2.96), not being omnivorous (RR 1.80, CI 1.08–3.00), and not smoking (RR 2.53, CI 1.06–6.07) was related to an increased chance of delivering at home compared to in a freestanding midwifery unit.

Of the 694 multiparas, most (68%) had previously given birth in a hospital and had a normal vaginal birth (95.7%). Most of the multiparas who delivered at home (69%) had previously given birth in a hospital, while about half of those who delivered in a freestanding midwifery unit (5.6%) had previously given birth at home (Table 2).

Almost all the deliveries (90.5%) were attended by two or more midwives, with a somewhat greater frequency in the freestanding midwifery units (Table 3).

The practice of being immersed in water during labor was chosen by 487 women (44.5%), and 47 actually gave birth while submerged in water. The most commonly used position in home deliveries was on all fours, while the chose position was squatting for deliveries in freestanding midwifery units [11% of women had nonpharmacological labor induction with castor oil, moxa, acupuncture, or membrane stripping]. In the third stage of labor, a quarter of women (24.2%) received a uterotonic agent (mainly oxytocin) more frequently in a freestanding midwifery unit. Women in the home birth group had a slightly higher rate of hemorrhage following birth compared with women who delivered in a freestanding midwifery unit. No third and one fourth degree perineal tear occurred in the study population, and only two episiotomies were performed. In the first week after delivery, 6 mothers (for break of stitches) and 19 newborns (5 suspected brachial plexus injury, 5 suspected infection, 2 hyperpyrexia, 2 jaundice, and 5 not well defined) were transferred to the hospital, all after delivering at home. None of the other characteristics related to birth or birth outcome that were evaluated differed between the two delivery settings.

The comparison of the characteristics of low-risk women giving birth out-of-hospital to those of women who planned hospital births showed that the place of residence, age, number of children, level of education, and marital status were variables affecting the probability of having an out-of-hospital birth (Table 4). Although not for all out-of-hospital births, the respective five matched in-hospital controls resided in the same area, and difference in the municipality did not affect the results.

In the conditional logistic regression model, the significant factors in choosing to give birth out-of-hospital were living in a large or medium city (OR 2.20; 1.75–2.77; OR 2.41; 1.93–3.02) and a secondary or higher level of education of the parents for each stratification (OR > 2). Being a primipara, on the other hand, was slightly significant (1.69;1.39–2.05) also in influencing the choice of giving birth in a freestanding midwifery unit. Sensitivity analysis on two separate models where dependent variables with numbers of missing values of all considered variables were added did not change the overall results of the study.

## 4. Discussion

The population enrolled in this first Italian study corresponds to 58.8% of the expected national deliveries over the study period and for all causes of out-of-hospital births. Considering that the population was selected, since only healthy women with low-risk pregnancies represented the investigated population, the findings are representative for this selected population and support the satisfying choice (also because it was shown to be safe) of giving birth out-of-hospital.

Being over 35 years old, multiparous, highly educated, married and/or cohabiting with a partner with a high level of education, and living in a small town were factors that increased the probability of having a birth out-of-hospital in Italy. Being a primipara increased the probability of delivering in a midwifery unit compared to at home. Findings reflect those from studies performed in other European countries, in the US, Canada, Japan, New Zealand, and Australia [14,26,27,28,29,30,31], although why the difference exists must still be understood. Multiparity was associated with out-of-hospital births. This is in line with previous studies [26,32,33], suggesting that the feeling of risk is greater for primiparas. Many factors are involved in women’s choice of place of birth and include cultural attitudes, religion, and peer and family standings, but a determinant that leads to choosing out-of-hospital delivery is also previous birth experience [32,33,34,35]. Moreover, giving birth at home involves celebration, togetherness, and ontological security [36]. These reasons are valid for comparable settings in countries with middle and high resource availability, while in countries with low resources, where out-of-hospital birth rates are high, poverty, access to hospitals, and lack of transportation to the nearest facility determine the choice [37,38]. In the present study, the proportion of transfer from home to hospital after planned out-of-hospital births is much lower than reported [39]. However, it is essential that a prompt transfer to hospital, in particular the emergency transfer, is guaranteed to all women (also those with low-risk pregnancies) if complications occur [40].

This study did not have the power to identify small differences in variables with low incidence and, unfortunately, as in similar countries [41], the out-of-hospital birth rate in Italy is low. The results can thus be indicative for countries with comparable services and societal organization. The strength of the study was that it acquired, for the first time, and via a rigorous data collection process, detailed information on Italian out-of-hospital births that were based on a formal, up-to-date, evidence-based assistance protocol, in a large population. The first national dataset was created and included details on women’s characteristics, pregnancy monitoring, labor, and birth and neonatal outcomes of interest. These data have contributed to the knowledge base by providing evidence comparing delivery settings. 

The comparative safety of different birth settings is widely debated. In the context of continuous comparison and evaluation, findings of the present study support that for women with low-risk pregnancies in high-income countries, planned place of birth appears to have little significant impact on adverse perinatal outcomes [42].

## 5. Conclusions

Out-of-hospital births in women with low-risk pregnancies is possible and can be a safe choice. It is an option that needs to be planned, monitored, regulated, and evaluated according to health care control systems in order to guarantee, as in hospitals, the safest and most effective care to a mother and her newborn(s).

## Figures and Tables

**Table 1 ijerph-17-02718-t001:** Maternal and partner characteristics by birth at home or in a midwifery unit.

Population in Study Characteristics	Delivered at Home (n = 848)	Delivered in a Freestanding Midwifery Unit (n = 251)	Overall (N = 1099)	RR at Home Birth (95% CI)	F (*p*-Value)
Maternal characteristics					
Age in years (mean ± SD), n (%)	34.0 ± 4.7	34.2 ± 4.6	34.0 ± 4.7		1.04 (0.71)
18–24	18 (2.1)	5 (2.0)	23 (2.1)	1.00 (0.81–1.25)	
25–34	436 (51.4)	125 (49.8)	561 (51.1)	Reference	
≥35	394 (46.5)	121 (48.2)	515 (46.8)	0.98 (0.92–1.05)	
Residential area, n (%)					
Large city	291 (37.3)	113 (51.1)	404 (40.4)	Reference	
Medium-size city	333 (42.7)	63 (28.5)	396 (39.6)	1.17 (1.08–1.26)	
Small town	155 (20.0)	45 (20.4)	200 (20.0)	1.08 (0.98–1.18)	
Missing	69	30	91		
Marital status, n (%)					
Married and/or cohabiting	729 (86.2)	222 (89.6)	951 (87.0)	Reference	
Other	119 (13.8)	29 (10.4)	148 (13.0)	1.07 (0.98–1.16)	
Number of children, n (%)					
First	257 (31.1)	113 (47.5)	370 (34.8)	Reference	
Second or more	569 (68.9)	125 (52.5)	694 (65.2)	1.18 (1.09–1.27)	
Missing	22	13	35		
First delivery >35 years, n (%)					
Yes	70 (8.3)	35 (14.7)	105 (9.9)	0.85 (0.74–0.97)	
No	756 (89.1)	203 (85.3)	959 (90.1)	Reference	
Missing	22	13	35		
Level of education, n (%)					
Primary	18 (2.1)	8 (3.2)	26 (2.3)	Reference	
Secondary	257 (30.3)	61 (24.3)	318 (28.9)	1.17 (0.90–1.52)	
Post-secondary	568 (67.0)	182 (72.5)	750 (68.2)	1.09 (0.84–1.42)	
Missing	5	-	5		
Occupation status before index birth, n (%)					
Working	658 (77.9)	206 (82.4)	864 (78.9)	Reference	
Not working	186 (22.1)	44 (17.6)	230 (21.1)	1.06 (0.99–1.14)	
Missing	4	1	5		
Annual income (€), n (%)					
<20.000	90 (13.2)	23 (10.9)	113 (12.6)	Reference	
20–29.000	229 (33.5)	58 (27.5)	287 (32.1)	1.00 (0.90–1.12)	
≥30.000	364 (53.3)	130 (61.6)	494 (55.3)	0.93 (0.83–1.03)	
Missing	165	40	205		
Food features, n (%)					
Omnivorous	665 (78.5)	201 (81.1)	866 (79.1)	Reference	
Other	182 (21.5)	47 (18.9)	229 (20.9)	1.04 (0.96–1.12)	
Missing	1	3	4		
Smoking, n (%)					
Yes	21 (2.5)	15 (6.1)	36 (3.3)	0.75 (0.57–0.99)	
No	817 (97.5)	234 (93.9)	1051 (96.7)	Reference	
Missing	10	2	12		
Partner characteristics					
Age in years (mean ± SD)	37.0 ± 6.0	37.5 ± 5.9	37.1 ± 6.0		1.01 (0.90)
Level of education, n (%)					
Primary	62 (7.4)	22 (7.4)	84 (7.4)	Reference	
Secondary	384 (46.1)	106 (46.1)	490 (46.1)	1.06 (0.93–1.22)	
Post-secondary	387 (46.5)	114 (46.5)	501 (46.5)	1.05 (0.91–1.20)	
Missing	15	9	24		
Occupation status, n (%)					
Working	816 (98.2)	242 (99.2)	1058 (98.4)	Reference	
Not working	15 (1.8)	2 (0.8)	17 (1.6)	1.14 (0.96–1.37)	
Missing	17	7	24		

n = values; % percentage; RR = Relative Risk; CI = confidence interval; SD = standard deviation; F = F-test; and Z = Z-test.

**Table 2 ijerph-17-02718-t002:** Data on prior pregnancies [n, (%)], by birth at home or in a midwifery unit.

Place of Delivery	Delivered at Home (n = 569)	Delivered in a Freestanding Midwifery Unit (n = 125)	Overall (N = 694)	RR at Home Birth (95% CI)	Z or F (*p*-Value)
Home	160 (28.2)	7 (5.6)	167 (24.1)	Reference	
Freestanding midwifery unit	16 (2.8)	39 (31.2)	55 (7.9)	0.30 (0.20–0.46)	
Hospital	392 (69.0)	79 (63.2)	471 (68.0)	0.87 (0.83–0.91)	
Missing	1	0	1		
Mode of delivery, n (%)					
Normal vaginal birth	546 (96.0)	118 (94.4)	664 (95.7)	Reference	
Operative delivery	23 (4.0)	7 (5.6)	30 (4.3)	0.93 (0.76–1.14)	

n = values; % percentage; RR = Relative Risk; CI = confidence interval; and SD = standard deviation.

**Table 3 ijerph-17-02718-t003:** Birth-related characteristics by birth at home or in a midwifery unit.

Characteristics	Delivered at Home (n = 848)	Delivered in a Freestanding Midwifery Unit (n = 251)	Overall (N = 1099)	RR at Home Birth (95% CI)	Z or F (*p*-Value)
Birth-related					
Number of midwives at delivery, n (%)					
1	92 (10.9)	12 (4.8)	104 (9.5)	1.16 (1.08–1.26)	
2 or more	753 (89.1)	238 (95.2)	991 (90.5)	Reference	
Gestational age, weeks (mean; SD)	39.6; 1.0	39.7;1.0	39.6; 1.0		Z = 0.89 (0.37)
Use of water in labor, n (%)					
Yes	367 (3.5)	130 (48.0)	487 (44.5)	Reference	
No	477 (56.5)	130 (52.0)	607 (55.5)	1.04 (0.98–1.11)	
Position in delivering, n (%)					
Lying down	106 (12.5)	54 (21.5)	160 (14.6)	0.96 (0.84–1.1)	
Squatting	204 (24.0)	92 (36.7)	296 (26.9)	Reference	
Kneeling	46 (5.4)	4 (1.6)	50 (4.5)	1.33 (1.19–1.49)	
On all fours	365 (43.0)	62 (24.7)	427 (38.9)	1.24 (1.14–1.35)	
On the side	72 (8.5)	34 (13.5)	106 (9.6)	0.99 (0.85–1.15)	
Others	55 (6.6)	5 (2.0)	60 (5.5)	1.33 (1.19–1.48)	
Induction of labor, n (%)					
Yes	76 (9.3)	33 (13.6)	109 (10.3)	0.89 (0.79–1.02)	
No	741 (90.7)	210 (86.4)	951 (98.7)	Reference	
Fetal vertex presentation, n (%)					
Occiput-anterior	816 (97.4)	238 (96.0)	1054 (97.0)	Reference	
Occiput-posterior	22 (2.6)	10 (4.0)	32 (3.0)	0.89 (0.70–1.12)	
Uterotonic agent use, n (%)					
Yes	173 (20.4)	93 (37.1)	266 (24.2)	0.80 (0.73–0.88)	
No	675 (79.6)	158 (62.9)	833 (75.8)	Reference	
Cord clamping, min (mean ± SD)	101.7 (± 125.1)	109.8 (± 86.1)	103.6 (± 117.2)		F = 2.11 (0.0001)
Lotus, n (%)					
Yes	298 (36.0)	67 (28.0)	365 (34.2)	1.08 (1.01–1.15)	
No	529 (64.0)	172 (72.0)	701 (65.8)	Reference	
Birthweight, g (mean ± SD)	3435.0 (± 451.1)	3413.3 (± 391.9)	3438.7 (± 402.2)		F = 1.32 (0.008)
Small for gestational age, n (%)					
Yes	111 (13.1)	30 (12.0)	141 (12.8)	1.02 (0.93–1.12)	
No	737 (86.9)	221 (88.0)	958 (87.2)	Reference	
Exclusive breastfeeding at 10 days, n (%)					
Yes	808 (97.4)	239 (95.6)	1047 (97.0)	Reference	
No	22 (2.6)	10 (4.4)	32 (3.0)	0.89 (0.70–1.13)	
Birth-outcomes					
Postpartum hemorrhage, n (%)					
≤500 mL	746 (88.5	227 (90.8)	973 (89.0)	Reference	
>500 mL	97 (11.7)	23 (9.2)	120 (11.0)	1.05 (0.96–1.16)	
Perineal tear, degree, n (%)					
No	417 (52.5)	123 (49.4)	594 (51.8)	Reference	
1st	278 (31.0)	91 (36.5)	369 (32.2)	0.98 (0.91–1.05)	
2nd	147 (16.4)	35 (14.1)	182 (15.9)	1.05 (0.96–1.14)	
3rd	-	-	-	-	
4th	1 (0.1)	-	1 (0.1)	-	
Mother’s postpartum hospitalization (within 1 week of delivery), n	5	1	6		
Newborn’s hospitalization (within 1 week of birth), n	16	3	19		

n = values; % percentage; RR = Relative Risk; CI = confidence interval; SD = standard deviation; F = F-test; and Z = Z-test.

**Table 4 ijerph-17-02718-t004:** Maternal and partner characteristics, by out-of-hospital and in hospital births.

Characteristics	Out-of-Hospital Birth (N = 1099)	Hospital Birth (N = 5495)	OR for Out-of-Hospital Birth (95% CI)	F (*p*-Value)
Maternal characteristics				
Age in years (mean ± SD), n (%)	34.0 ± 4.7	33.2 ± 5.0		F = 1.15 (0.003)
18–24	23 (2.1)	304 (5.5)	0.38 (0.25–0.59)	
25–34	561 (51.1)	2830 (51.5)	Reference	
≥35	515 (46.8)	2361 (43.0)	1.10 (0.97–1.26)	
Residential area, n (%)				
Large city	404 (40.4)	2045 (37.7)	Reference	
Medium-size city	396 (39.6)	2727 (50.2)	0.74 (0.63–0.85)	
Small town	200 (20.0)	657 (12.1)	1.54(1.27–1.87)	
Missing	99	66		
Marital status, n (%)				
Married and/or cohabiting	951 (87.0)	3696 (67.3)	Reference	
Other	148 (13.0)	1798 (32.7)	0.32(0.27–0.38)	
Missing		1		
Number of children, n (%)				
First	370 (34.8)	1398 (25.4)	Reference	
Second or more	694 (65.2)	4097 (74.6)	0.64 (0.56–0.74)	
Missing	35			
First delivery >35 year of age, n (%)				
Yes	105 (9.9)	274 (5.0)	2.01 (1.59–2.55)	
No	959 (90.1)	5221 (95.0)	Reference	
Missing	35			
Level of education, n (%)				
Primary	26 (2.3)	1212 (22.1)	Reference	
Secondary	318 (28.9)	2436 (44.3)	6.09 (4.06–9.13)	
Post-secondary	750 (68.2)	1847 (33.6)	18.93 (12.72–28.17)	
Missing	5			
Occupation status before index birth, n (%)				
Working	864 (78.9)	4414 (80.3)	Reference	
Not working	230 (21.1)	1081 (19.7)	1.09 (0.93–1.28)	
Missing	5			
Partner characteristics				
Level of education, n (%)				
Primary	84 (7.4)	1678 (31.1)	Reference	
Secondary	490 (46.1)	2476 (45.9)	3.95 (3.11–5.02)	
Post-secondary	501 (46.5)	1243 (23.0)	8.05 (6.32–10.26)	
Missing	24	98		
Occupation status before index birth, n (%)				
Working	1058 (98.4)	5466 (98.5)	Reference	
Not working	17 (1.6)	29 (0.5)	3.03 (1.66–5.53)	
Missing	24			

n = values; % percentage; OR = Odd Ratio; CI = confidence interval; SD = standard deviation; and F = F-test.

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
