# Peer review of "Low-Risk Planned Out-of-Hospital Births: Characteristics and Perinatal Outcomes in Different Italian Birth Settings"

_ijerph, 2020, doi:10.3390/ijerph17082718_

Round 1
Reviewer 1 Report
congratulations on a good, clear, effective and useful research study- which has numerous strengths, I will, however, restrict myself to focussing on the minor issues which need clarification/correction:
Line 40- has citations [6-9] after the word however but it does not seem clear why there is a citation at this point?
Line 49 where you refer to all causes (in the context of home births) does this mean planned/unplanned deliveries? If so i think it would be better to say so rather than all causes
Line 53 should be: 'varriables which predict them'- rather than 'varribales that can characterize them'.
Line 67- suggest 'The National Association of Out-of-Hospital Birth Midwives' is changed to: 'The National Association of Out-of-Hospital Midwives'
Line 76- 'Women cared for by the midwives of association...' should read: 'Women cared for by the out of hospital midwives association...'
Line 124: 'for parity, age...' should read 'for parity of: age...'
Table 2 & 3 need legends- CI= Confidence interval, values are N and % etc.
Table 2- in the delivered at home column you report the number of hospital births... is this because they were actually 'intended delivery at home- but then went into hospital for some reason? this needs clarifying
Line 166 add OR inside the brackets.
Author Response
congratulations on a good, clear, effective and useful research study- which has numerous strengths, I will, however, restrict myself to focussing on the minor issues which need clarification/correction:
Line 40- has citations [6-9] after the word however but it does not seem clear why there is a citation at this point?
Some words were missing. Text has been corrected.
Line 49 where you refer to all causes (in the context of home births) does this mean planned/unplanned deliveries? If so i think it would be better to say so rather than all causes
It has done.
Line 53 should be: 'varriables which predict them'- rather than 'varribales that can characterize them'.
It has done.
Line 67- suggest 'The National Association of Out-of-Hospital Birth Midwives' is changed to: 'The National Association of Out-of-Hospital Midwives'
It has done.
Line 76- 'Women cared for by the midwives of association...' should read: 'Women cared for by the out of hospital midwives association...'
It has done.
Line 124: 'for parity, age...' should read 'for parity of: age...'
It has been corrected.
Table 2 & 3 need legends- CI= Confidence interval, values are N and % etc.
It has done.
Table 2- in the delivered at home column you report the number of hospital births... is this because they were actually 'intended delivery at home- but then went into hospital for some reason? this needs clarifying
Table 2 refers to previous births by the women in study. Thus, they they may have given birth in the hospital earlier.
Line 166 add OR inside the brackets.
It has done.
Reviewer 2 Report
This article deals with an interesting topic.
1) Most of all, I would like to ask authors to align their hypotheses more closely with existing literature in the Introduction section. Why do authors think readers may be interested in this topic? What is not known in the literature? There are some systematic reviews authors may want to cite; including https://www.cochranelibrary.com/cdsr/doi/10.1002/14651858.CD000352.pub2/abstract and https://www.ncbi.nlm.nih.gov/pubmed/29408739.
2) For the sample selection, if I'm not mistaken, authors said that the control group consists of hospital births "in the Lombardy Region", while the out-of-hospital birth samples are from the whole country - is it right? If so, authors should put more thoughts on the comparability between the two groups - preferably by using only out-of-hospital births in the Lombardy. If not, please clarify in the Methods section.
3) I would see the results not only as risk ratios but also as absolute (marginal) risk differences. Authors may present additional results in Appendix.
4) Was there a time trends in out-of-hospital births in Italy? Has it been increasing or decreasing over time? And, why is that (for example, what's the health insurer's reimbursement policy regarding hospital vs. home births)? Is there any qualitative studies on the reasons why Italian mothers choosing out-of-hospital births?
Author Response
This article deals with an interesting topic.
- Most of all, I would like to ask authors to align their hypotheses more closely with existing literature in the Introduction section. Why do authors think readers may be interested in this topic? What is not known in the literature? There are some systematic reviews authors may want to cite; including https://www.cochranelibrary.com/cdsr/doi/10.1002/14651858.CD000352.pub2/abstract and https://www.ncbi.nlm.nih.gov/pubmed/29408739.
In the introduction it was reported and cited the wide inter-county rate of out-of-hospital births, as a few of the potential sources, among which the characteristics of national health and social organization. We also underlined and cited that this is an area with little undertaken research.
Review https://www.cochranelibrary.com/cdsr/doi/10.1002/14651858.CD000352.pub2/abstract was the ref. 10.
Review https://www.ncbi.nlm.nih.gov/pubmed/29408739 has been added (new ref. 13).
- For the sample selection, if I'm not mistaken, authors said that the control group consists of hospital births "in the Lombardy Region", while the out-of-hospital birth samples are from the whole country - is it right? If so, authors should put more thoughts on the comparability between the two groups - preferably by using only. If not, please clarify in the Methods section.
In the Methods section it has been made explicit. No difference was found considering only out-of-hospital births in the Lombardy.
- I would see the results not only as risk ratios but also as absolute (marginal) risk differences. may present additional results in Appendix.
What requested has been added in Appendix. Information was reported in the Methods section.
- Was there a time trends in out-of-hospital births in Italy? Has it been increasing or decreasing over time? And, why is that (for example, what's the health insurer's reimbursement policy regarding hospital vs. home births)? Is there any qualitative studies on the reasons why Italian mothers choosing out-of-hospital births?
To our knowledge this is the first formal study which will be published in the scientific literature about out-of-hospital birth in Italy. Over the past decade the prevalence of out-of-hospital births is constant, although in a few regions a partial reimbursement has been recognized.
Round 2
Reviewer 2 Report
- Authors might have misunderstood my comment on matching. I pointed out that case and control sample should come from the same region - the manuscript still reads that the case sample is selected nationwide, but the control data set is only from Lombardy region. As we can reasonably think that choice of delivery place would be affected by differences in region-level healthcare system or social belief, for comparability, the case group also should be those living in the Lombardy region. Though I do not believe that this manipulation changes the results much, but the current matching is scientifically nonsense.
- The phrase in the Abstract "The risks of hospitalization for women with low-risk pregnancies, and their babies, within the first week of delivery were very low (6 of 1099 mothers; 19 of 1099 neonates)." is way too strong and downstream. There is a huge risk of selection into out-of-hospital birth in that low-risk mothers are more likely choose it. Thus, authors' results indicate almost nothing about the postnatal outcomes with this setting. I'd recommend to delete this phrase.
- "There are no official detailed statistics in Italy about these births, but the rate is reported to be around 0.04-0.01%" in the page 2 -> isn't it "0.04-0.1%" or "0.004-0.01%"?
- I've noticed some typos including "Variations in the level of integration into the health care system across different settings is a" ("Variation in..." would be right) in page 2. Put more efforts on English editing.
Author Response
Dear Editor,
Below are our responses to the Reviewer’s comments. A reviewer who should know the purposes of the reviewing a paper, that it is an activity between peers, and that words make sense.
- Authors might have misunderstood my comment on matching. I pointed out that case and control sample should come from the same region - the manuscript still reads that the case sample is selected nationwide, but the control data set is only from Lombardy region. As we can reasonably think that choice of delivery place would be affected by differences in region-level healthcare system or social belief, for comparability, the case group also should be those living in the Lombardy region. Though I do not believe that this manipulation changes the results much, but the current matching is scientifically nonsense.
If the residential area affects the choice of delivering out-of-hospital or in-hospital was not one of the aims of the study. Adequate stratification by geographic setting needs larger samples. As for other studies, this is the first in Italy, results are reported as national figure.
However, for control when possible, municipality was considered. As reported in the result section, no different was found comparing population living in or out Lombardy.
- The phrase in the Abstract "The risks of hospitalization for women with low-risk pregnancies, and their babies, within the first week of delivery were very low (6 of 1099 mothers; 19 of 1099 neonates)." is way too strong and downstream. There is a huge risk of selection into out-of-hospital birth in that low-risk mothers are more likely choose it. Thus, authors' results indicate almost nothing about the postnatal outcomes with this setting. I'd recommend to delete this phrase.
The task of an observational study is to observe and describe. The sentence has been reworded.
- "There are no official detailed statistics in Italy about these births, but the rate is reported to be around 0.04-0.01%" in the page 2 -> isn't it "0.04-0.1%" or "0.004-0.01%"?
It has been corrected
- I've noticed some typos including "Variations in the level of integration into the health care system across different settings is a" ("Variation in..." would be right) in page 2. Put more efforts on English editing.
It has been corrected
